# Novel Approach to Improving Specialist Access in Underserved Populations with Suspicious Oral Lesions

**James Nguyen** [1]**, Susan Yang** [1]**, Anastasya Melnikova** [1]**, Mary Abouakl** [1]**, Kairong Lin** [1]**, Thair Takesh** [1]**,
Cherie Wink** [1]**, Anh Le** [2]**, Diana Messadi** [3]**, Kathryn Osann** [4] **and Petra Wilder-Smith** [1,*]

[1] Beckman Laser Institute and Medical Clinic, University of California Irvine School of Medicine, Irvine, CA 92612, USA

[2] Department of Oral and Maxillofacial Surgery/Pharmacology, University of Pennsylvania, Philadelphia, PA 19104, USA

[3] UCLA School of Dentistry, University of California Los Angeles, Los Angeles, CA 90095, USA

[4] School of Medicine, University of California Irvine, Irvine, CA 92617, USA

[*] Correspondence: pwsmith@hs.uci.edu; Tel.: +1-949-824-7632

**Abstract:** Late detection and specialist referral result in poor oral cancer outcomes globally. High-risk LRMU populations usually do not have access to oral medicine specialists, a specialty of dentistry, whose expertise includes the identification, treatment, and management of oral cancers. To overcome this access barrier, there is an urgent need for novel, low-cost tele-health approaches to expand specialist access to low-resource, remote and underserved individuals. The goal of this study was to compare the diagnostic accuracy of remote versus in-person specialist visits using a novel, low-cost telehealth platform consisting of a smartphone-based, remote intraoral camera and custom software application. A total of 189 subjects with suspicious oral lesions requiring biopsy (per the standard of care) were recruited and consented. Each subject was examined, and risk factors were recorded twice: once by an on-site specialist, and again by an offsite specialist. A novel, low-cost, smartphone-based intraoral camera paired with a custom software application were utilized to perform synchronous remote video/still imaging and risk factor assessment by the off-site specialist. Biopsies were performed at a later date following specialist recommendations. The study's results indicated that on-site specialist diagnosis showed high sensitivity (94%) and moderate specificity (72%) when compared to histological diagnosis, which did not significantly differ from the accuracy of remote specialist telediagnosis (sensitivity: 95%; specificity: 84%). These preliminary findings suggest that remote specialist visits utilizing a novel, low-cost, smartphone-based telehealth tool may improve specialist access for low-resource, remote and underserved individuals with suspicious oral lesions.

**Keywords:** oral cancer; screening; specialist access; telehealth; underserved populations

## 1. Introduction

Worldwide, 650,000 incident cases and 223,000 deaths from oral cancers (OCs) are reported each year. In the US, 54,000 OC cases and 13,500 deaths occur annually [1–3]. Low resource, minority and underserved (LRMU) populations have the highest rates of OC in the US and a higher prevalence of oral potentially malignant lesions (OPMLs) [4–9], a heterogeneous group of lesions with varying malignant transformation rates [10]. OPMLs affect 2% of the world's population, with a mean malignant transformation rate of 7.9% [11,12], which increases up to 24% in high-risk lesions [13].

Early diagnosis is the primary determinant of OC outcomes, with 5-year survival approximating 20% in individuals diagnosed after metastasis, but 80% in those diagnosed at an early stage [14]. An increase in the time to treatment by as little as 2 months significantly increases the risk of death [12], yet the mean time to specialist diagnosis in LRMU populations is 6 months [15,16]. Therefore, early diagnosis and minimal time to treatment are crucial to improving outcomes, with the most impactful OC prevention intervention

being the early detection and management of OPMLs. Individuals from LRMU populations are diagnosed later and have considerably poorer outcomes for OC than others [5–8,15–19]. Disparities in access to care, timely diagnosis and management are major drivers for these poor outcomes [20]. The COVID-19 pandemic has worsened this situation: At one local community dental clinic, compliance with referral for OC risk dropped from 50% to 3% after the onset of the COVID-19 pandemic [21]. Moreover, while telemedicine in many specialties has expanded exponentially during the COVID-19 pandemic and studies recognize the role of telemedicine to improve access to healthcare [22–27], telehealth visits with oral medicine and oral surgery (OM/OS) specialists are offered by very few specialist centers. When telehealth visits are available, audio communications and oral video examinations are typically both conducted on the patient's smartphone, which is logistically difficult and does not permit the effective inspection of up to 40% of the oral cavity [28].

A recent pilot study found that compliance with OM/OS specialist referral improved from 46% to 86% in 60 LRMU patients with suspicious oral lesions if the remote specialist option was provided [29]. Since OC outcomes correlate strongly with disease stage at treatment initiation [14], improving specialist access will translate directly into better OC outcomes and management in high-risk LRMU populations [21].

During the COVID-19 pandemic, the decline in vital dental services contributed to reduced inpatient and outpatient services. In India, the average number of oral cancer-related outpatient and inpatient visits was reduced by 63% and 51.4%, respectively, between the time period from April 2019 to March 2020 and the period from April 2020 to June 2020 [30]. Delays in diagnosis are associated with increased tumor size and metastases at diagnosis. Among those undergoing tumor resection in the same study, 81% had a significantly advanced tumor stage and composite stage, which is 20% more than during the pre-COVID-19 pandemic era [30].

COVID-19-pandemic-related delays in dental care were more impactful in LRMU populations, lower- and middle-income (LMIC) countries and disadvantaged groups such as those >75 years old. In the UK, the prevalence of oral lesions that had advanced to actual oral cancers was higher in these vulnerable populations during the COVID-19 pandemic [31]. The number of specialist referrals for lesions suspicious for oral cancer was significantly reduced by 70.4% in England during the suspension of routine dental exams during the initial lockdown [31]. In Italy, the >75-years-old age group had not recovered from the initial pandemic-related decline in oral specialist visits by the end of 2020 [32].

The objective of this study was to compare the accuracy of OM specialist in-person diagnosis versus synchronous remote examination by the same OM specialist utilizing a subject-operated telehealth platform that consisted of a novel, low cost, smartphone-based intraoral camera paired with a custom software application. This study was performed in LRMU individuals with oral lesions suspicious for oral pre-cancer or cancer lesions. The long-term goal of the study is to overcome the existing inability of oral healthcare specialists to conduct effective, virtual oral examinations due to critical limitations in remote visualization and clinical evaluation [33].

## 2. Materials and Methods

This project was conducted in full compliance with University of California at Irvine's IRB-approved protocol #2002-2805. Written informed consent was obtained from all the subjects involved in the study.

### 2.1. Overview

The diagnostic performance of an oral medicine specialist during an in-person visit versus a remote synchronous visit using a telehealth platform prototype consisting of a novel, low-cost, smartphone-based intraoral camera and custom software application was assessed in patients with intraoral lesions suspicious for OPML or oral cancer. Biopsies were performed as the standard of care that provided the gold standard for assessing diagnostic accuracy.

### 2.2. Subjects

Subjects aged 18 years and above who had been referred to University of California, Irvine clinics with oral lesions suspicious of OPML or oral cancer were recruited, and written informed consent was collected. Patients who were under 18 years of age or had undergone previous treatment targeting the oral cavity in the form of chemotherapy, radiotherapy, surgery or medication were excluded from the study.

### 2.3. Protocol

This randomized study had two arms. For the first study arm, the subjects received a routine, standard-of-care in-person oral medicine specialist examination and diagnosis, including visual examination, palpation, as well as the documentation of symptoms and risk factors. In the second study arm, the subjects were examined in a separate examination room, where they were provided with the intraoral camera platform and a 5-minute training session in its use. Then, the same oral medicine specialist performed a full visual examination remotely from an adjacent room using the intraoral camera which was operated by the subjects following voice instructions by the specialist that were transmitted over the smartphone component of the intraoral camera platform. The same risk factor compilation as during the in-person visit was also collected remotely by the specialist. Based on this information, the specialist recorded a second diagnosis. Subjects were randomized 1:1 (randomizer.com, accessed on 3 October 2020) as to the sequence of remote versus in-person specialist visits.

### 2.4. Intraoral Camera Platform

A prototype, HIPAA-compliant, remote intraoral camera was used for this study (Figure 1). A prototype software application was installed on the remote specialist's computer and on the smartphone that was connected to the intraoral camera via a USB port. At the time of the telehealth visit, the patient accepted the specialist's call by clicking on the telehealth icon shown on the phone screen. This initiated the connection between phone and intraoral camera, which immediately allowed the specialist to perform a synchronous, high-resolution visual inspection, and it also provided the ability to record images or videos during the examination. The specialist verbally guided the patient through the phone to operate the intraoral camera as needed. Infection control was implemented through the use of custom disposable sheaths. Symptoms and risk factors were documented by the specialist via the software application on their computer.

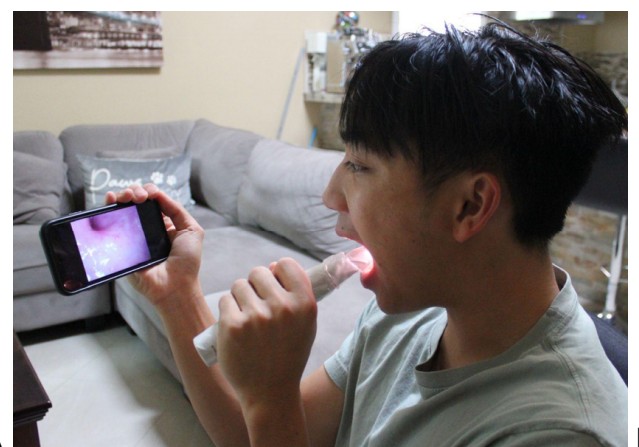 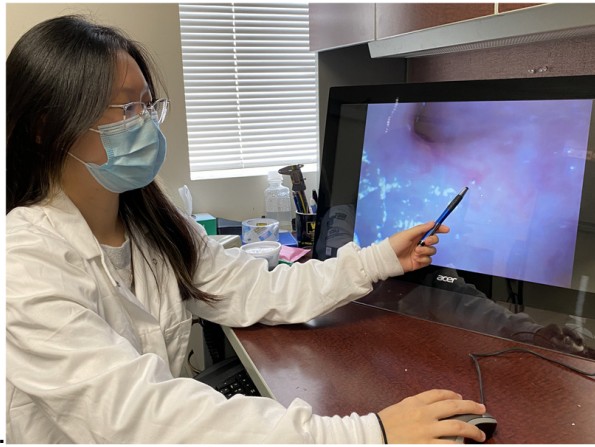

**Figure 1.** (**A**) Patient self-operating the intraoral camera, which is connected to the smartphone via USB cable. (**B**). Clinician remotely performs synchronous intraoral inspection.

*2.5. Statistical Analysis*

Sensitivity, specificity, agreement, false positive rate, false negative rate and positive and negative predictive values were estimated from the observed results. Standard errors (SE) and 95% confidence intervals were calculated for all the rates. A level of $p < 0.05$ was used to indicate statistical significance.

## 3. Results

*3.1. Subjects*

In total, 189 subjects were recruited and provided written consent, including 78 females and 111 males. Subjects' ages ranged from 42 to 89 years old, with a mean age of 68 years old. A total of 52 subjects identified as Hispanic; 48 subjects identified as White; 41 identified as Asian; 27 identified as more than one race; 19 identified as African American; and 2 identified as Pacific Islander (Table 1). There was a total of 17 final diagnosis categories for patients (Table 2). All the subjects completed the study in full compliance with the protocol.

**Table 1.** Patients' profile of gender, race, and age.

| Gender | Race | Age |
|:---:|:---:|:---:|
| 78 Female | 52 Hispanic | 42–89 years old |
| 111 Male | 48 White | Mean of 68 years old |
| | 41 Asian | |
| | 27 more than one race | |
| | 19 African American | |
| | 2 Pacific Islander | |

**Table 2.** Categories of patients' final diagnosis and number of cases in each category.

| Category | Cases |
|:---:|:---:|
| Vascular | 2 |
| Pigmented | 3 |
| Traumatic | 4 |
| Infectious | 5 |
| Periodontal or Dental | 6 |
| Anatomic | 2 |
| Candida | 4 |
| Keratotic | 17 |
| Lichen Planus | 11 |
| Hyperplastic | 19 |
| Nicotinic | 33 |
| OPML | 64 |
| OSCC | 19 |

*3.2. Diagnostic Accuracy vs. Histopathology*

The in-person versus remote specialist diagnostic performances were very similar (Table 3). In-person diagnoses provided 92.6% agreement, 94.9% sensitivity, and 69.2% specificity versus histopathology, which is considered the gold standard. Remote specialist diagnoses provided 92.1% accuracy, 94.8% sensitivity, and 62.5% specificity versus histopathology. The examination sequence (in-person visit vs. remote visit) did not significantly affect the diagnostic accuracy vs. the gold standard.

**Table 3.** Diagnostic performance vs. histopathology for in-person and remote specialists.

|  |  | Value | SE | Lower CI | Upper CI |
|---|---|---|---|---|---|
| In-Person | Sensitivity | 0.949 | 0.017 | 0.916 | 0.981 |
|  | Specificity | 0.692 | 0.128 | 0.443 | 0.942 |
|  | False Positive Rate | 0.308 | 0.128 | 0.058 | 0.557 |
|  | False Negative Rate | 0.057 | 0.017 | 0.023 | 0.091 |
|  | Agreement with Histopathology | 0.926 | 0.019 | 0.889 | 0.963 |
| Remote | Sensitivity | 0.948 | 0.017 | 0.915 | 0.981 |
|  | Specificity | 0.625 | 0.121 | 0.389 | 0.861 |
|  | False Positive Rate | 0.375 | 0.121 | 0.139 | 0.611 |
|  | False Negative Rate | 0.052 | 0.017 | 0.019 | 0.085 |
|  | Agreement with Histopathology | 0.921 | 0.020 | 0.882 | 0.959 |

## 4. Discussion

The purpose of this study was to compare the diagnostic accuracy of oral medicine specialists in individuals with oral lesions suspicious of OPML or oral cancer. Specialist diagnoses were performed twice: once within an in-person setting, and again within a synchronous remote setting. The reasoning behind this study rests on the observation that a leading cause of delay in accessing the pathway to care is patient reluctance or the inability to seek specialist care. This, in turn, is associated with poor compliance with referral to specialist care [34]. Reasons for reluctance include costs, unfamiliarity, language, distance, and fear/avoidance [35,36]. One large study reported that on average, patients travel twice as far to access a specialty oral health clinic compared to visits to their referring dentist or doctor. Physical distance constitutes an additional barrier to compliance with a specialist referral [37]. Moreover, dental clinicians tend to "observe" suspect lesions rather than refer the patient immediately, resulting in an additional mean delay in referral of 3 months [34]. Together with lack of regular dental care for many of the LRMU populations, these circumstances leave high-risk populations as the most susceptible to late referrals and often encounter access barriers to the pathway of care. Thus, there exists an urgent need for novel approaches to overcome barriers to specialist access in LRMU populations.

The United States has a limited number of clinicians who specialize in oral medicine and are trained in oral cancer early detection, and even fewer practice in high-risk countries such as India [38–41]. These specialists tend to be clustered in academic centers, further limiting access [42]. According to one study which evaluated the distribution of oral medicine specialists in 20 US states in 2015, 46% of oral medicine specialists practiced at a dental school faculty practice, while 31% practiced in a hospital setting [43]. This study also reported that patients visited a mean of 2.2 clinicians before consulting with an oral medicine specialist, and there was a mean delay of 16.8 months between the first symptoms and the specialist consultation. These findings are dishearteningly similar to those published in a study 15 years earlier, in which the number of clinicians consulted prior to specialist visits also averaged 2.2, and the mean delay measured at 15 months [43,44].

Teledentistry represents a promising approach to improving specialist access and reducing diagnostic delays, especially in individuals with oral cancer risk. Clinicians are becoming increasingly aware of telehealth as a whole [45,46], yet very few dentists or dental specialists have implemented the practice. A survey of 2767 dentists in the United States reported that only 23% of dentists used teledentistry or virtual platforms [47]. A literature review reported emerging evidence in support of teledentistry [48], due to factors such as potential cost-minimization, avoiding the need for in-person attendance at distant specialist centers and innovative adjunct technology such as the addition of telecytology to the

telediagnosis of oral lesions [45,48,49]. However, current remote consultations are typically facilitated using a smartphone, which is unable to access all areas of the mouth, has focus and lighting challenges that hinder effective visualization, and requires imaging to be conducted while concurrently speaking on the very same phone. The prototype telehealth platform that was evaluated in this pilot study addresses these barriers in multiple ways, the foremost of which is achieving similar diagnostic accuracy compared to the results of an in-person specialist visit, while also providing an effective, low-cost, easy-to-use intraoral imaging probe, with functionalities that include the ability to focus, as well as optimized lighting for intraoral use. Moreover, the telehealth platform is compatible with almost any type and level of smartphone, including very simple Android phones with only basic capabilities. Many of our telehealth studies in LRMU countries use basic Android phones which are widespread in rural and remote areas of India.

We recognize that the ultimate diagnostic criterion for oral cancer risk is surgical biopsy and histopathology, which cannot be performed remotely. However, the proposed approach does address several challenges that serve as critical barriers to moving individuals into the pathway for care sooner. Firstly, it would serve to overcome bottlenecks in access to overburdened oral cancer specialists due to the over-referral of healthy individuals related to the poor specificity of oral cancer screening approaches by allowing specialists to perform a direct evaluation of the oral cavity and of risk factors. Secondly, allowing the patient to meet, see and speak with the specialist will reduce anxiety and build trust, paving a pathway to better compliance with an in-person visit—if needed—and entry into the pathway of care.

**5. Conclusions**

This study demonstrated that a novel low-cost, smartphone-based telehealth platform consisting of an intraoral camera and custom software application can be utilized to perform synchronous remote specialist intraoral examinations that provide similar levels of diagnostic accuracy as in-person diagnosis. The next steps include expanding the study design to involve multiple oral cancer specialists, the performance of remote intraoral examinations with patients in a non-clinic setting to simulate actual conditions during intended use, and as well as the optimization of user interfaces.

**Author Contributions:** J.N.: Data acquisition, interpretation and analysis, drafting article. K.L., S.Y., A.M. and M.A.: data acquisition and interpretation. K.O.: study design, statistical analysis and interpretation; drafting article. T.T. and C.W.: study conception and design, data interpretation. D.M. and A.L.: study conception and design, data analysis and interpretation, drafting and revising article critically, final approval of version to be submitted. P.W.-S.: study conception and design, data acquisition, data analysis and interpretation, drafting and revising article critically, final approval of version to be submitted. All authors have read and agreed to the published version of the manuscript.

**Funding:** TRDRP T31IR1825; NCRR NCATS NIH UL1 TR0001414; NIH UH3EBO22623; NIH RO1DE030682; University of California, Irvine Undergraduate Research Opportunities Program.

**Institutional Review Board Statement:** The study was conducted according to the guidelines of the Declaration of Helsinki and approved by the Institutional Review Board (or Ethics Committee) of the University of California, Irvine, under protocol #2002-2805.

**Informed Consent Statement:** Informed consent was obtained from all subjects involved in the study.

**Data Availability Statement:** Data supporting reported results may be available upon request.

**Conflicts of Interest:** The authors declare no conflict of interest.

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
