# Peer review of "Novel Approach to Improving Specialist Access in Underserved Populations with Suspicious Oral Lesions"

_curroncol, doi:10.3390/curroncol30010080_

Round 1
Reviewer 1 Report
How does this study use, can be extrapolated to real LRMU patients? They still have to visit the specialist for a biopsy and need the special probe with good quality phones which they don't have in the first place being LRMU.
Referencing style is not standardized (some journals use the short form, some use full name)
References #23 and #24 are incomplete.
Author Response
Dear Madam/Sir,
Many thanks for transmitting to us the reviewers’ critiques of our manuscript entitled “Novel Approach to Improving Specialist Access in Underserved Populations with Suspicious Oral Lesions”. We greatly appreciate the time and effort involved, both by yourself and your editorial team, as well as the esteemed reviewers.
The feedback from the reviewers was very helpful and serve to improve and strengthen the paper. We have incorporated the reviewers’ suggestions into the revised manuscript. The amendments are detailed below.
Review #1:
- How does this study use, can be extrapolated to real LRMU patients? They still have to visit the specialist for a biopsy and need the special probe with good quality phones which they don't have in the first place being LRMU.
To clarify this important point, we have added the following explanation to the “Discussion”:
We recognize that the ultimate diagnostic criterion for oral cancer risk is surgical biopsy and histopathology, which cannot be performed remotely. However, the proposed approach does address several challenges that serve as critical barriers to moving individuals into the pathway for care sooner. Firstly, it would serve to overcome bottlenecks in access to overburdened oral cancer specialists due to over-referral of healthy individuals related to the poor specificity of oral cancer screening approaches by allowing specialists to perform a direct evaluation of the oral cavity and of risk factors. Secondly, allowing the patient to meet, see and speak with the specialist will reduce anxiety and build trust, paving a pathway to better compliance with an in-person visit – if needed- and entry into the pathway of care.
- Referencing style is not standardized (some journals use the short form, some use full name)
We apologize for this oversight and have standardized the referencing style.
- References #23 and #24 are incomplete.
We apologize for this oversight. We have added the missing information.
We hope that these revisions and clarifications have addressed the reviewers’ questions and critiques.
Sincerely,

Reviewer 2 Report
The authors described "Novel Approach to Improving Specialist Access in Underserved Populations with Suspicious Oral Lesions" in a case series study. This study compared the diagnostic accuracy of oral medicine specialists in individuals with oral lesions suspicious of OPML or oral cancer. This topic is very important and informative for potential readers especially in the era of covid19 pandemic. I have some recommendations to improve this manuscript.
1. There were few references. So, please add the references related to telemedicines (e.g. teledermatology).
2. Who were specialists? Dentists? Head and neck surgeons?
3. I would like to know the patients' profiles including final diagnosis.
Author Response
Dear Madam/Sir,
Many thanks for transmitting to us the reviewers’ critiques of our manuscript entitled “Novel Approach to Improving Specialist Access in Underserved Populations with Suspicious Oral Lesions”. We greatly appreciate the time and effort involved, both by yourself and your editorial team, as well as the esteemed reviewers.
The feedback from the reviewers was very helpful and serve to improve and strengthen the paper. We have incorporated the reviewers’ suggestions into the revised manuscript. The amendments are detailed below.
Review #2:
- There were few references. So, please add the references related to telemedicines (e.g., teledermatology).
We have included additional references to telemedicine and teledermatology to provide a broader view of the field of telehealth
- Who were specialists? Dentists? Head and neck surgeons?
Dentists specialized in oral medicine- we have revised the manuscript to present this important information more clearly under section 2.3 of the protocol.
- I would like to know the patients' profiles including final diagnosis.
We have added a table providing this information in the “Subjects” paragraph of the “Results” section of the manuscript.
We hope that these revisions and clarifications have addressed the reviewers’ questions and critiques.
Sincerely,

Round 2
Reviewer 2 Report
The authors revised the manuscript precisely.